# Corrosion Behavior of Shot-Peened Ti6Al4V Alloy Produced via Pressure-Assisted Sintering

Egemen Avcu [1,2,*] , Eray Abakay [3] , Yasemin Yıldıran Avcu [1] , Emirhan Çalım [1] , İdris Gökalp [4] ,
Eleftherios Iakovakis [5] , Funda Gül Koç [4] , Ridvan Yamanoglu [4] , Akın Akıncı [3] and Mert Guney [6,7,*]

1   Department of Mechanical Engineering, Kocaeli University, Kocaeli 41001, Turkey;
    yaseminyildiran89@gmail.com (Y.Y.A.); emircalim@gmail.com (E.Ç.)
2   Ford Otosan Ihsaniye Automotive Vocational School, Kocaeli University, Kocaeli 41650, Turkey
3   Department of Metallurgy and Materials Engineering, Sakarya University, Sakarya 54050, Turkey;
    eabakay@sakarya.edu.tr (E.A.); akinci@sakarya.edu.tr (A.A.)
4   Department of Metallurgy and Materials Engineering, Kocaeli University, Kocaeli 41001, Turkey;
    idris.gokalp@kocaeli.edu.tr (İ.G.); fundagulkoc@gmail.com (F.G.K.); ryamanoglu@gmail.com (R.Y.)
5   Department of Mechanical Aerospace and Civil Engineering, The University of Manchester,
    Manchester M13 9PL, UK; eleftherios.iakovakis@manchester.ac.uk
6   Department of Civil and Environmental Engineering, Nazarbayev University, Astana 010000, Kazakhstan
7   The Environment & Resource Efficiency Cluster (EREC), Nazarbayev University, Astana 010000, Kazakhstan
*   Correspondence: avcuegemen@gmail.com (E.A.); mert.guney@nu.edu.kz (M.G.);
    Tel.: +90-26-2435-6167 (E.A.); +7-71-7270-4553 (M.G.)

**Abstract:** For the first time, the present study investigates the corrosion, surface, and subsurface properties of a shot-peened Ti6Al4V powder metallurgical alloy produced via pressure-assisted sintering. Shot peening yielded a fine-grained microstructure beneath the surface down to 100 microns, showing that it caused severe plastic deformation. XRD analysis revealed that the sizes of the crystallites in unpeened and shot-peened Ti6Al4V alloy samples were 48.59 nm and 27.26 nm, respectively, indicating a substantial reduction in crystallite size with shot peening. Cross-sectional hardness maps of shot-peened samples showed a work-hardened surface layer, indicating a ~17% increase in near-surface hardness relative to unpeened samples. Three-dimensional surface topographies showed that shot peening yielded uniform peaks and valleys, with a maximum peak height of 4.83 μm and depth of 6.56 μm. With shot peening, the corrosion potential shifted from −0.386 V to −0.175 V, showing that the passive layer developed faster and was more stable than the unpeened sample, improving corrosion resistance. As determined via XRD analysis, the increased grain refinement (i.e., the number of grain boundaries) and the subsequent accumulation of $TiO_2$ and $Al_5Ti_3V_2$ compounds through shot peening also suggested the effective formation of a protective passive layer. As demonstrated via electrochemical impedance spectroscopy, the formation of this passive film improved the corrosion resistance of the alloy. The findings will likely advance surface engineering and corrosion research, enabling safer and more productive shot peening in corrosion-critical applications.

**Keywords:** hardness; grain refinement; powder metallurgy; shot peening; titanium alloys; topography

## 1. Introduction

Titanium (Ti) alloys are widely employed in biomedical, chemical, automotive, and aerospace applications due to their high specific strength, corrosion resistance, non-toxicity, excellent biocompatibility, and fatigue properties [1–4]. More specifically, due to its superior corrosion resistance, the Ti6Al4V alloy has found widespread use in marine and biological applications [5–7]. The passive layer with a thickness of 4–6 nm that forms on the surface of the alloy is responsible for its high corrosion resistance [8]. In accordance with the nature of the powder metallurgy (PM) method, the corrosion properties of the Ti6Al4V alloy differ from those produced via conventional casting procedures. The porosities within the microstructure and especially the compositional differences at the grain boundaries affect

the corrosion properties of the Ti6Al4V alloy processed via PM [9]. For instance, a recent study [9] suggested that the corrosion resistance of PM titanium alloys may differ from that of conventionally cast titanium alloys (i.e., ingot metallurgy) on account of their porous characteristics and varied microstructural futures. This suggests that further investigation into the corrosion behavior of PM titanium alloys is required to elucidate the distinction in corrosion resistance between PM and conventional casting titanium alloys [9].

To date, several surface treatments have been adopted to improve the undesirable outcomes and corrosion behavior of PM-processed Ti alloys [10,11]. Additionally, peening-based methods such as laser peening [12–14], ultrasonic peening [15,16] and shot peening [1,17–20] have recently been applied to both conventional and powder metallurgical Ti alloys to enhance their mechanical, tribological, biological, and corrosion properties [12,14–20]. More importantly, it has been recently demonstrated that peening methods, such as laser peening [21], ultrasonic shot peening [15,22], and pneumatic shot peening [23], can enhance the corrosion resistance of the Ti6Al4V alloy [15,21–23].

Thus far, some contradicting results have been reported regarding the corrosion behavior of shot-peened Ti6Al4V alloys. While some claim that the peening process increases corrosion resistance as a result of the modified surface structure [24,25], others assert the opposite [26]. According to Zhang et al. [15], proper ultrasonic shot-peening (USSP) treatment improves the corrosion resistance of a selective laser-melted Ti6Al4V alloy in aqueous 3.5% NaCl medium, attributed to the increased microstructural defects (e.g., dislocation density) due to USSP, promoting passive film formation on the peened surface. Vella et al. [18] applied a duplex surface treatment (shot peening (SP) and physical vapor deposition (PVD)) to an additively produced Ti6Al4V alloy, demonstrating that the treatment does not improve the corrosion resistance of the alloy. It was suggested that SP increases surface roughness and forms an inhomogeneous and unstable layer [18]. Kumar et al. [27] showed that the corrosion resistance of the Ti13Nb13Zr alloy increases with USSP due to grain refinement and compressive stress on the alloy's surface. In summary, the formation of a passive layer in titanium alloys seems beneficial for enhancing corrosion resistance, whereas the modification of the microstructural and surface properties (e.g., grain refinement, dislocation accumulation, surface roughness, etc.) with shot peening affects passive layer formation, thereby influencing corrosion behavior. While the literature highlights the importance of these parameters' impact on corrosion behavior, it is impossible to determine which parameter has a dominant effect on shot-peening-induced corrosion behavior.

To the authors' knowledge, this is the first study where the corrosion behavior of a Ti6Al4V alloy produced via pressure-assisted sintering is thoroughly researched. The surface, microstructure, and mechanical properties of the Ti6Al4V alloy after shot peening were investigated to understand the effect of shot peening on the alloy's corrosion properties. The present study aims to contribute to the limited literature on the effects of shot peening on the corrosion behavior of titanium alloys, whose corrosion properties are essential for their use in a variety of engineering applications such as biomedical (i.e., orthopedic and dental implants) [28–31], marine [9] (i.e., propeller shafts and transmission components) [18], heat exchange (i.e., heat exchanger tubes) [32,33], and aerospace (i.e., aero-engines and gas turbines) [34].

## 2. Materials and Methods

Figure 1 presents a summary of this investigation. Using powder metallurgy, Ti6Al4V alloy samples were produced, which were then shot-peened. The surface characteristics of unpeened and shot-peened samples were analyzed using optical profilometry and scanning electron microscopy (SEM). In addition, the cross-sectional microstructure and hardness maps of shot-peened samples were analyzed to determine the effect of shot peening on microstructural and mechanical properties. Finally, potentiodynamic polarization and electrochemical impedance spectroscopy were used to evaluate the electrochemical corrosion properties in an aqueous 0.01 M phosphate-buffered saline medium.

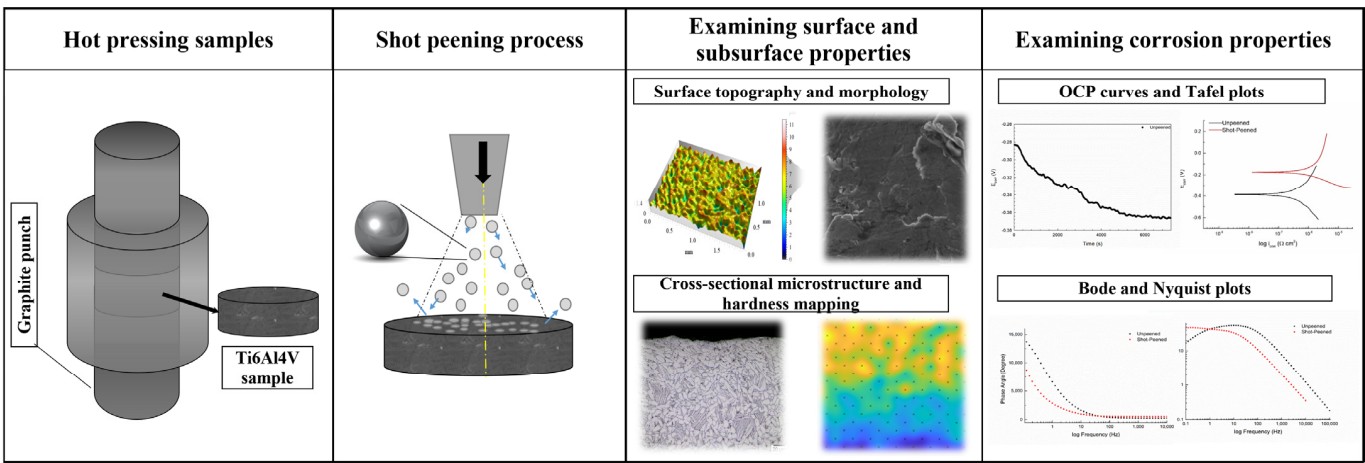

**Figure 1.** Outline of the present study.

## 2.1. Processing Ti6Al4V Samples

Utilizing a hot press (DIEX, Shanghai, China), cylindrical button powder metallurgical Ti6Al4V samples (height: 4 mm, diameter: 20 mm) were manufactured from Ti6Al4V powder (size range: 20–70 μm, having a spherical morphology (Figure 2)). Graphite paper was used to prevent the powder from sticking to the graphite mold during the production process. The heights of the die punches were adjusted uniformly to provide a uniform temperature distribution during sintering. To avoid powder oxidation, the sintering procedure was conducted in a vacuum ($10^{-4}$ mbar). The samples were sintered for 30 min at 950 °C and 14 kN. They were then allowed to cool to room temperature within the hot press. Before shot peening, the samples were ground using 320–2000 grits.

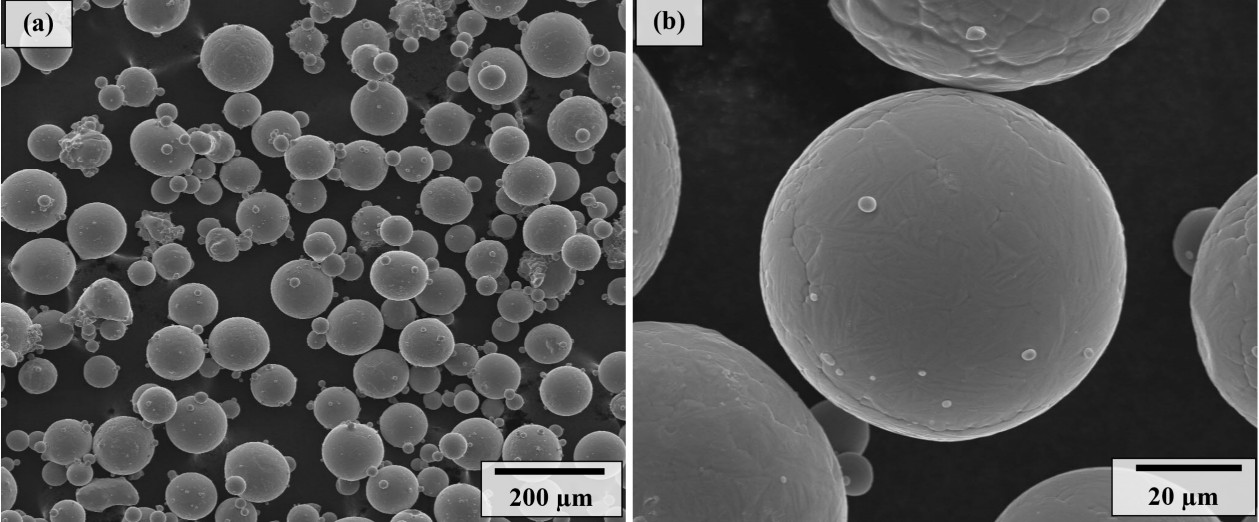

**Figure 2.** (**a**) Low- and (**b**) high-magnification secondary electron SEM images of Ti6Al4V powder particles, indicating a spherical and homogeneous shape.

## 2.2. Shot Peening

The surfaces of Ti6Al4V alloy samples were shot-peened using a custom-designed, fully automated shot-peening system; shot peening's operating parameters were chosen depending on previous research [1,17,35,36]. Fundamentally, shot parameters were carefully set to maximize shot peening's benefits (i.e., surface hardness and microstructural characteristics) without microcrack formation or excessive surface roughness. Ti6Al4V samples were peened with stainless steel shots (size range: 600 to 1000 μm (Figure 3))

accelerated with compressed air (7 bar) for 150 s. After shot peening, the samples were cleaned in an ultrasonic bath containing alcohol to remove particles and dust from the samples' surface.

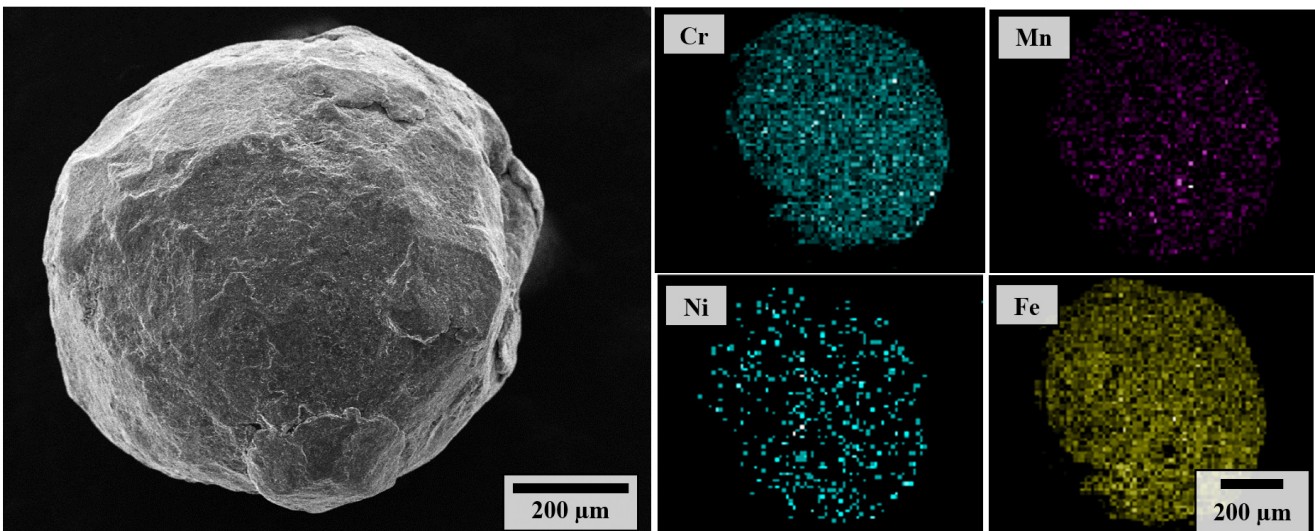

**Figure 3.** Secondary-electron SEM image and EDS analysis of stainless-steel shots used for peening.

### 2.3. Characterisation of Surface Properties

Shot-peened surfaces were scanned at a high resolution using a 3D optical profilometer (Huvitz, Gyeonggi-do, South Korea). Mountains® 9 (Digital Surf, Besançon, France) was then used to assess the 3D surface topographies and surface roughness characteristics. SEM in secondary-electron and backscattered modes was used to analyze the surface morphology of shot-peened materials (Jeol JSM-6060, Akishima, Japan).

### 2.4. Characterisation of Microstructural Properties

For hardness examination, cross-sections of shot-peened samples were metallographically prepared through the following procedure: the samples were ground with 320–2000 mesh grits, polished with 6, 3, and 1 μm diamond suspensions, and etched with a Kroll solution (2 mL HF, 6 mL $HNO_3$, 92 mL $H_2O$). Then, SEM images of the cross-sectional microstructure were captured in backscatter mode using the previously described SEM.

Quantitative phase analyses of unpeened and shot-peened powder metallurgical Ti6Al4V alloys were carried out using X-ray diffraction (XRD). Analyses were performed with a diffractometer (Rikagu, DMax 2200, Akishima-Shi, Japan) using Cu-K$\alpha$ radiation between 20° and 90°, with a step size of 2°. In addition, crystallite size was determined using the XRD analysis results. Details of the method can be found elsewhere [37].

### 2.5. Characterisation of Mechanical Properties

For hardness evaluation, metallographically prepared cross-sections of the shot-peened samples (as described in the previous subsection) were used to reveal the variation in micro-hardness in the shot-peening-affected region. In accordance with ASTM E348-17, Vickers microhardness tests with three repetitions were performed using 0.2 kgf for 10 s and 0.1 mm spacing. The hardness impressions were produced within a large region (rectangular grid of $900 \times 540$ μm$^2$, located around 30 μm beneath the surface). The hardness maps were then visualized using a modified version of a MATLAB® tool code, as reported before [17,38,39].

### 2.6. Characterisation of Corrosion Properties

Untreated and shot-peened powder metallurgical Ti6Al4V samples were subjected to electrochemical corrosion tests in an aqueous 0.01 M phosphate-buffered saline (PBS)

solution. Analytical-grade PBS (Sigma-Aldrich, Burlington, MA, USA) was dissolved in distilled water to form the solution (one tablet per 200 mL of water). The experiments were conducted in a typical three-electrode cell with untreated and shot-peened samples serving as the working electrode (WE), graphite serving as the counter electrode (CE), and calomel serving as the reference electrode (RE). The sample areas of the circular-shaped working and counter electrodes were 0.28 and 2.54 (cm$^2$), respectively. Prior to conducting the potentiodynamic polarization and EIS tests, the open circuit potential of untreated and shot-peened samples was measured for 7200 and 1800 s, respectively, until the corrosion potential stabilized. Under atmospheric conditions, potentiodynamic polarization measurements were conducted with a sweep rate of 0.1 across a potential range of $-0.25$ V to 0.25 V. The Tafel extrapolation was executed with the Echem Analyst software developed by Gamry Instruments, Warminster, PA, USA. The initial frequency for EIS tests was 10 kHz, and the ultimate frequency was 10 MHz. The amplitude of the sine wave was 10.0 mV AC, and the frequency was 10 Hz per decade. The experiments involving both approaches were conducted in triplicate.

## 3. Results

### 3.1. Surface Properties of Shot-Peened Ti6Al4V Alloy

Two-dimensional surface maps and 3D surface topographies reveal that shot peening forms uniformly distributed peaks and valleys on the surface due to the plastic deformation caused by the repeated impact of steel shots (Figure 4a,b), with a maximum peak height and depth of 4.83 μm and 6.56 μm, respectively (Figure 4b). According to the roughness profile (Figure 4c) extracted from the shot-peened surface, 1.5 μm-high peaks and valleys formed after shot peening, compared to the unpeened sample with a mirror-polished surface (0.03 μm arithmetical surface roughness).

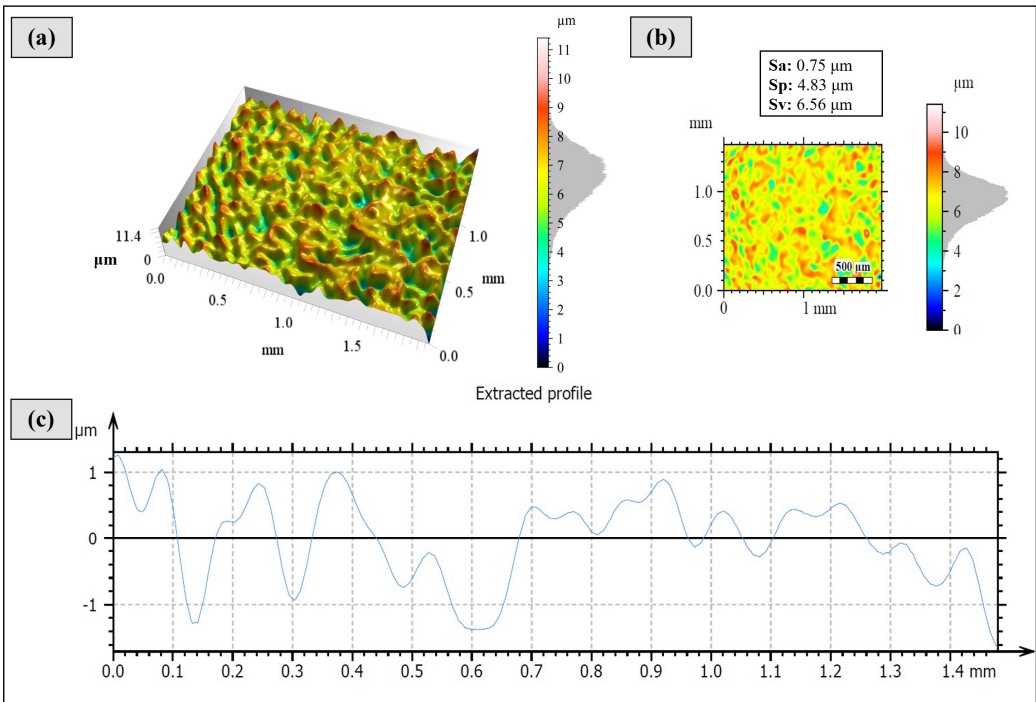

**Figure 4.** (**a**) Three-dimensional surface topography, (**b**) 2D roughness map, and (**c**) roughness profile of the shot-peened samples.

SEM micrographs of the shot-peened surfaces clearly show the effects of shot peening on the surface morphology, indicating severe plastic deformation on the surface (Figure 5a–d). The severe plastic deformation that occurs due to shot peening leads to the formation of peaks and valleys, as described by the topography maps (Figure 4a,b). However, the titanium

alloy's limited deformability due to its hexagonal lattice crystallographic structure prevents the formation of smooth and deep craters, whilst smearing and folding are still visible on the surface (Figure 5c). The folding mechanisms are the result of excessive plastic deformation, while the repeated impact of the shots causes more smearing and the overlap of the folded surfaces (Figure 5d).

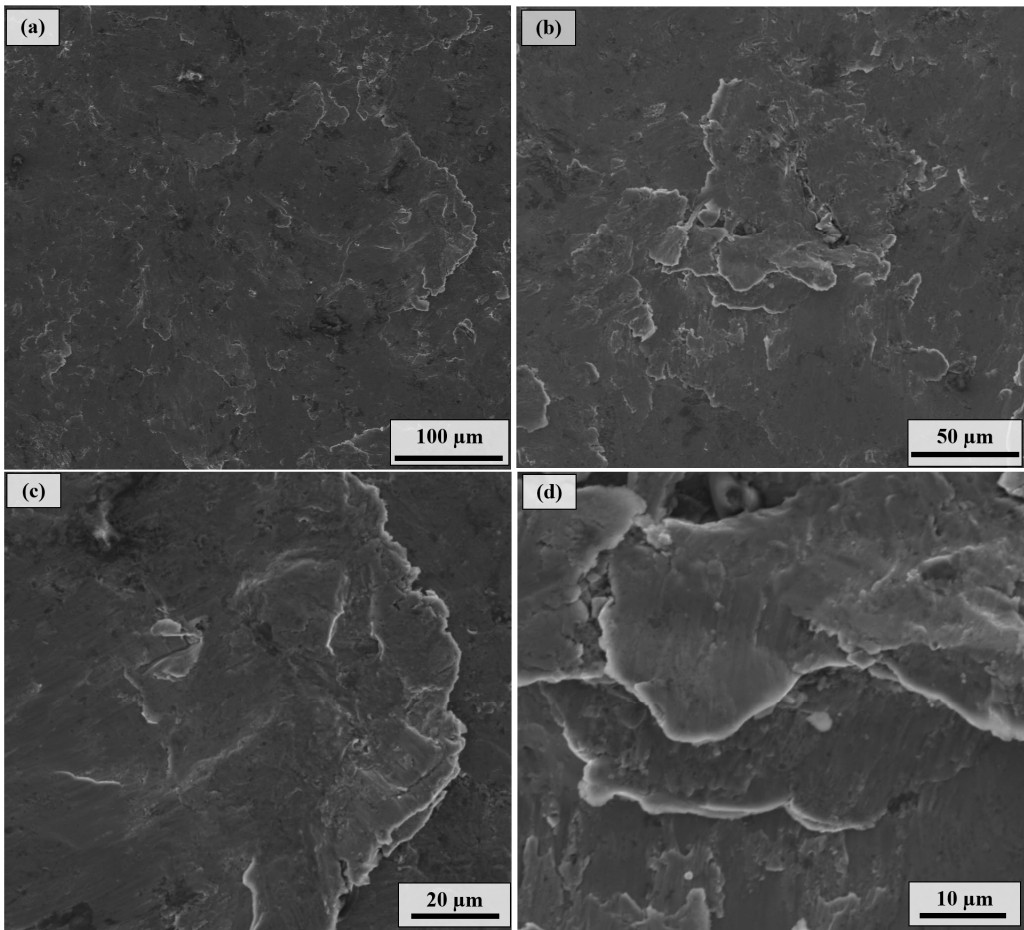

**Figure 5.** Low-magnification (**a**,**b**) and high-magnification (**c**,**d**) secondary-electron SEM images of the surface of shot-peened Ti6Al4V.

### 3.2. Corrosion Properties of Shot-Peened Ti6Al4V Alloy

The analysis of the corrosion potential ($E_{corr}$) of the unpeened and shot-peened Ti6Al4V alloy in 0.01 M aqueous PBS solution (Figure 6a,b) shows that the corrosion potential for the untreated sample ($E_{corr}$) decreased for 5000 s and then remained stable until the end of the test. The SP sample had a different $E_{corr}$ variation than the unpeened sample. The potential dropped for 600 s, with a smaller decline than in the absence of the operation (about 0.1 V for no operation and 0.005 V for SP). Up to around 400 s, the corrosion potential indicated a threefold increase, followed by a rapid reduction. This was a result of the sample's rough and porous nature. The drop in potential lasted around 600 s, after which the increase stopped and stayed constant until the completion of the measurement.

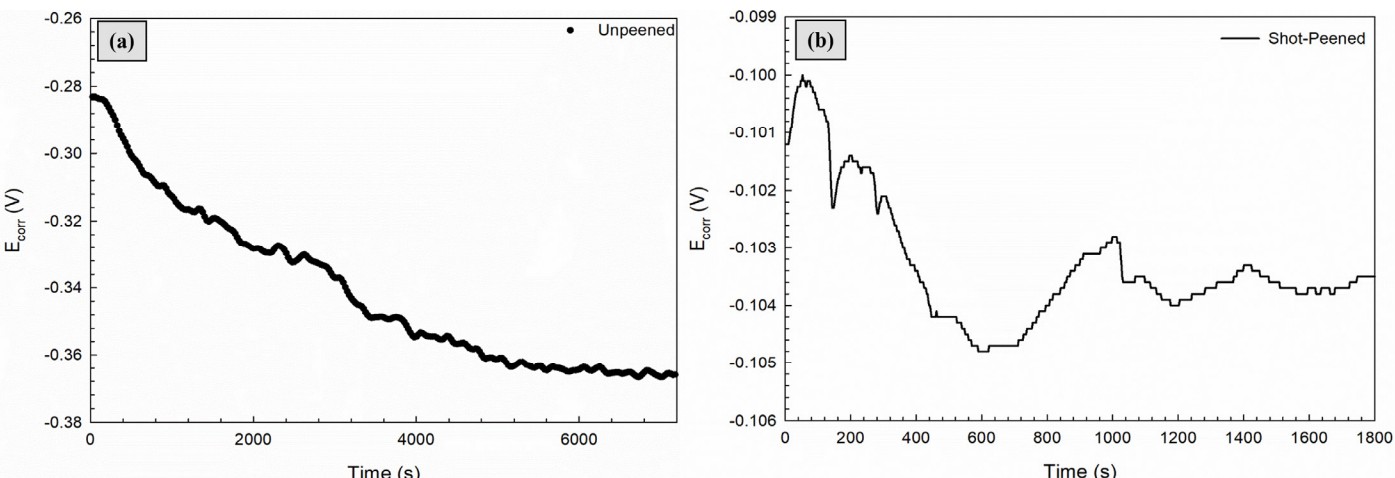

**Figure 6.** OCP curves of (**a**) unpeened and (**b**) shot-peened Ti6Al4V alloys.

In Figure 7, the potentiodynamic polarization curves of the unpeened and shot-peened Ti6Al4V alloys in 0.01 M phosphate-buffered saline solution are displayed. Tafel extrapolation yielded $E_{corr}$ and $i_{corr}$ values of $-0.356$ V and 1.73 μA, respectively, for the unpeened sample and $-0.175$ V and 2 μA, respectively, for the shot-peened sample. Table 1 shows Tafel extrapolation results for untreated and shot-peened samples. According to the Tafel extrapolation results, the corrosion rate is lower (2.050 mpy for the unpeened sample and 2.375 mpy for the shot-peened sample) for the unpeened sample. According to Stem–Geary equation, the corrosion rate is mostly related to the corrosion current density ($i_{corr}$) [28], where lower $i_{corr}$ values correspond to higher corrosion rates.

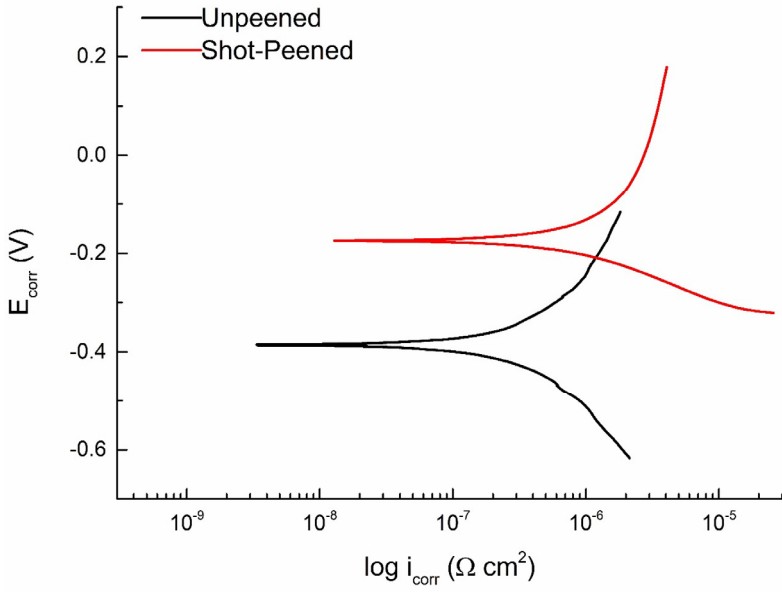

**Figure 7.** Potentiodynamic polarization plots of the untreated and shot-peened Ti6Al4V alloys.

**Table 1.** Tafel extrapolation results for the untreated and shot-peened samples.

|  | Untreated | Shot-Peened |
|---|---|---|
| $\beta_a$ | 1.439 V/decade | 1.084 V/decade |
| $\beta_c$ | $877.3 \times 10^{-3}$ V/decade | $182.8 \times 10^{-3}$ V/decade |
| $i_{corr}$ | 1.730 μA | 2.000 μA |
| $E_{corr}$ | $-386.0$ mV | $-175.0$ mV |
| $R_p$ | 2.050 mpy | 2.375 mpy |

Using Bode and Nyquist curves, the electrochemical impedance spectroscopy test results of the unpeened and shot-peened Ti6Al4V samples were investigated. Figure 8 displays Bode curves for the samples. The unpeened and shot-peened samples resulted in the formation of a Bode phase angle plot with a single peak.

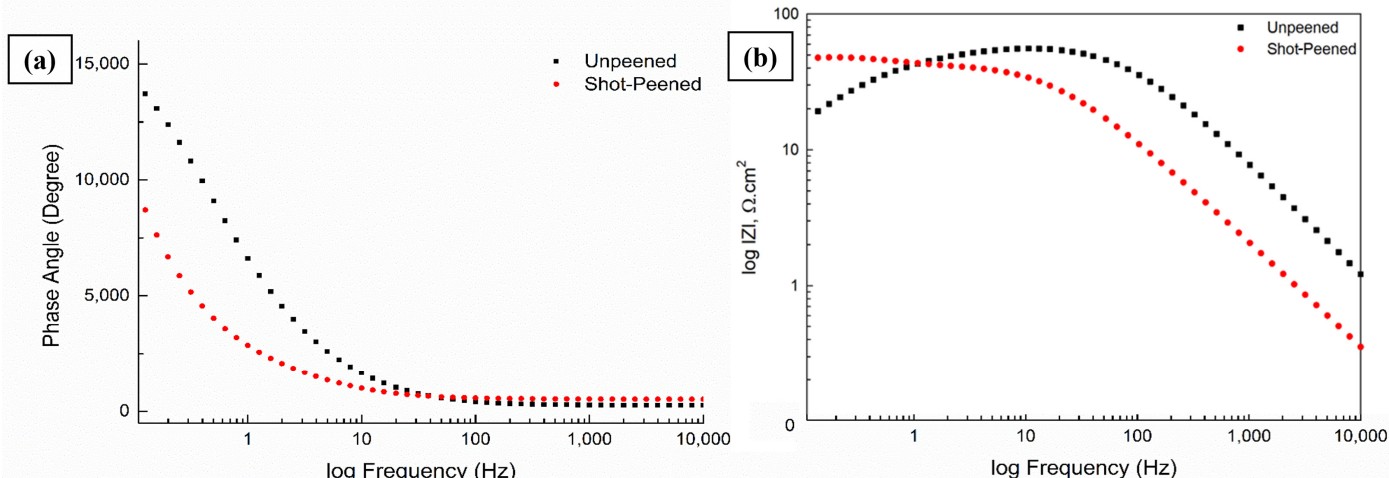

**Figure 8.** (**a**) Bode phase and (**b**) Bode magnitude plot diagrams of the unpeened and shot-peened Ti6Al4V alloys.

Figure 9 displays Nyquist diagrams of the unpeened and shot-peened Ti6Al4V alloys in 0.01 M phosphate-buffered saline solution. The unpeened sample resulted in a single semicircular arc. Consistent with the Bode curves, the development of a single semicircle shows that a single dissolving process is successful. The two semicircles illustrate that corrosion is caused by two distinct chemical processes. In addition, the dimensions of the semicircle provide information regarding the corrosion rate. Consequently, a rise in the diameter of the semicircle signifies a reduction in the corrosion rate.

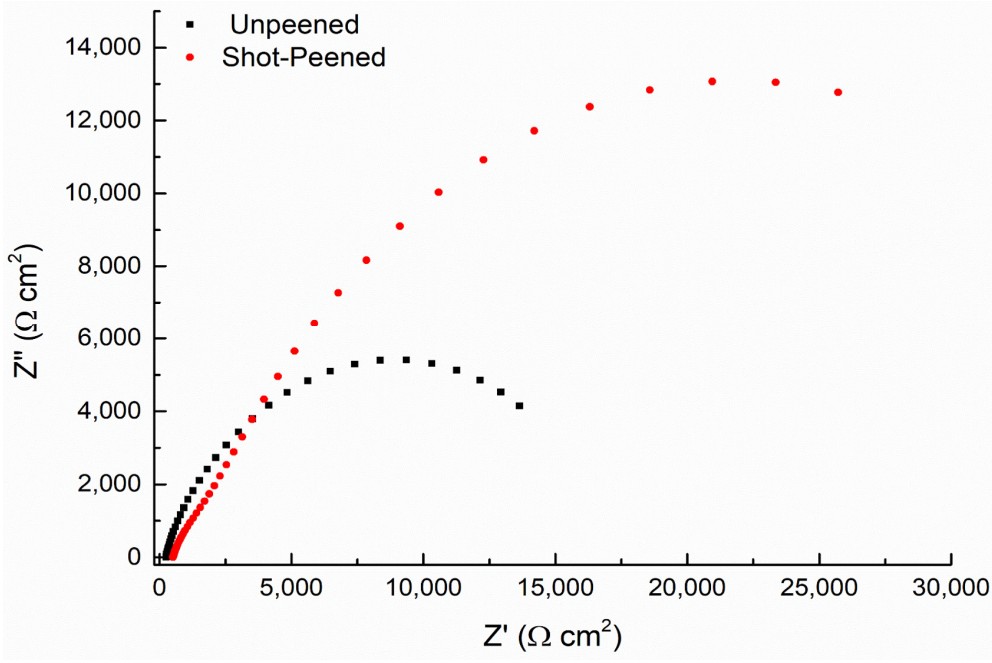

**Figure 9.** Nyquist plot diagrams of the unpeened and shot-peened Ti6Al4V alloys.

The equivalent circuit derived from EIS data for the unpeened and shot-peened powder metallurgical Ti6Al4V samples is illustrated in Figure 10. In the given circuit, Rs

denotes the solution's resistance, while Rp and Rb represent the resistances of the bulk and porous layers of the passive layer, respectively. Furthermore, the capacitances denoted as Qp and Qb correspond to the porous and bulk layers, respectively. Further information regarding these elements is available elsewhere. [30]. Comparable findings for the Ti6Al4V alloy were acquired in alternative research [40]. Table 2 gives the impedance parameters for unpeened and shot-peened samples. The shot-peened sample exhibits much higher Rp, Rb, and Rs values in comparison to the unpeened sample, suggesting that the passive film provides substantial protection [27]. Additionally, the shot-peened layer has a n value of 1, which is nearly ideal for the bulk layer [22].

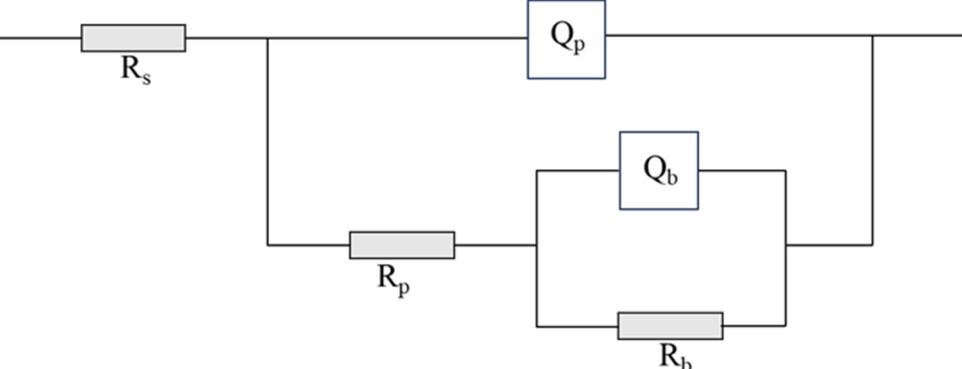

**Figure 10.** Equivalent circuit formed using the EIS results of unpeened and shot-peened powder metallurgical Ti6Al4V samples.

**Table 2.** Impedance parameters for unpeened and shot-peened samples.

|  | $R_s$ | $R_b$ | $C_b$ | $n_1$ | $R_p$ | $C_p$ | $n_2$ |
|---|---|---|---|---|---|---|---|
| Unpeened | 251.9 | $1.65 \times 10^3$ | $2.84 \times 10^{-5}$ | $7.62 \times 10^{-1}$ | $1.41 \times 10^4$ | $3.40 \times 10^{-9}$ | $6.84 \times 10^{-1}$ |
| Shot-Peened | 510.1 | $1.66 \times 10^4$ | $4.22 \times 10^{-6}$ | 1 | $4.32 \times 10^4$ | $1.19 \times 10^{-4}$ | $5.98 \times 10^{-1}$ |

## 4. Discussion

Shot peening had a distinct impact on the surface properties of the Ti6Al4V sample (roughness, topography, and morphology) (Figures 4 and 5). As previously explained and depicted [1,17,35], the plastic deformation caused by the impact of shots altered the surface characteristics as a function of the deformation mechanisms (such as folding, ridge removal, crater formation, etc.) [35]. These modifications not only led to an increase in surface roughness (Figure 4), but also changed the microstructural and mechanical characteristics of the surface and subsurface of the shot-peened Ti6Al4V samples. These changes notably affected the alloy's corrosion behavior, as discussed below.

Figure 11 depicts the hardness map derived from the subsurface region of the shot-peened Ti6Al4V alloy. Approximately 30 μm from the surface, the greatest hardness value was measured as 445 HV0.2. While the hardness values of the material continue to increase up to a depth of 210 μm, beyond this depth, hardness returns to the levels of the unpeened sample. The increase in hardness may be attributed to the grain refinement generated by the impact of the shots on the surface during shot peening (Figure 12) [1,17,35]. Grain refinement in the subsurface reduced dislocation movements and resulted in strain hardening (Hall–Petch mechanism) in the material [1,19,20,35]. Figure 12a,b show the cross-sectional microstructure of the shot-peened samples. Grain refinement occurred up to a depth of around 80–90 μm (Figure 12). These results correlate well with the previously discussed hardness map (Figure 11), since the subsurface regions with the finest grains demonstrate the highest improvement in hardness. Furthermore, valleys with a depth of up to 10 μm were found in cross-sectional investigations (Figure 12b), agreeing well with the surface topographies (Figure 4a) and indicating the induced plastic deformation on

the surface and subsurface during shot peening. In addition to grain refining and strain hardening, the samples became rougher. The absence of surface and subsurface microcracks following shot peening indicates that the peening parameters were appropriately tuned.

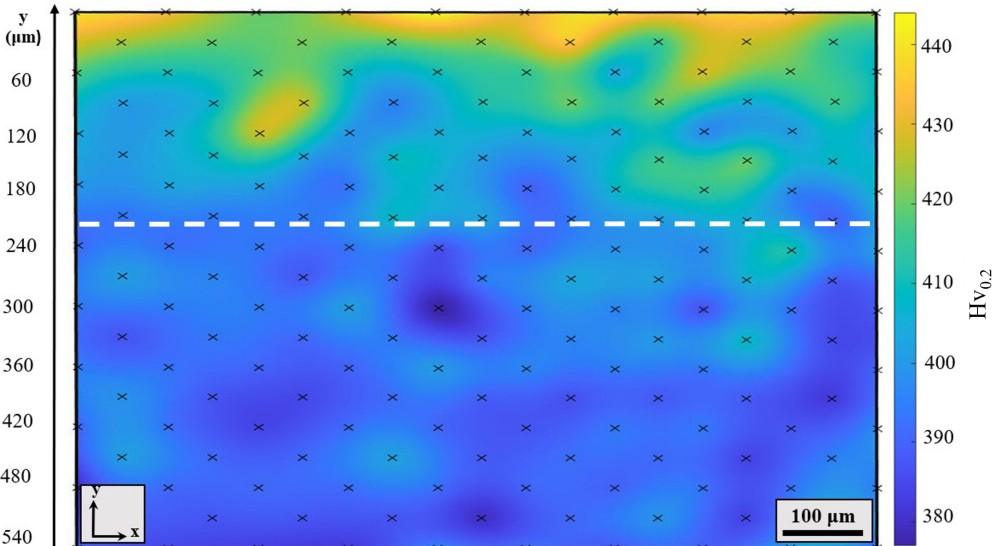

**Figure 11.** Cross-sectional hardness map of the shot-peened Ti6Al4V sample, illustrating surface/subsurface hardening of the material with shot peening. Markings (×) indicate the position of hardness imprints; the dashed line illustrates the depth of the hardened layer, which is roughly 210 μm.

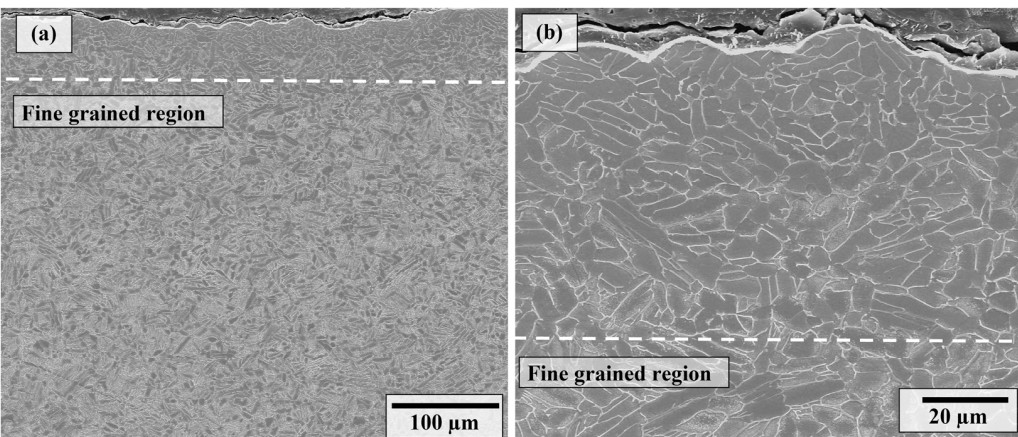

**Figure 12.** Cross-sectional backscattered SEM images of the shot-peened Ti6Al4V sample at (**a**) low and (**b**) high magnification.

To provide a more comprehensive understanding of the grain refinement and crystalline structure of the shot-peened samples, we here present XRD analyses of the unpeened and shot-peened powder metallurgical Ti6Al4V alloy samples (Figure 13). After shot peening, the intensities of the XRD peaks were altered, and the peaks shifted to the left and became broader. It has been reported that the broadening and shifting of the peaks occur due to grain refinement during shot peening [20]. The XRD analysis revealed that the sizes of the crystallites in the unpeened and shot-peened Ti6Al4V alloy samples were 48.59 nm and 27.26 nm, respectively. These values indicate that shot peening causes a substantial reduction in crystallite size and leads to a significant grain refinement at the near surface, consistent with previous studies that have reported similar results [41–45]. In addition, XRD investigations revealed that the predominant phase in the structure of the unpeened powder metallurgical Ti6Al4V is the α-Ti phase, with the β-Ti phase also being present as

a minor phase. These findings are consistent with the results of XRD analyses conducted on powder metallurgical Ti6Al4V alloys in prior research [46]. In addition to the α-Ti and β-Ti phases, $TiO_2$ and $Al_5Ti_3V_2$ compounds were detected after shot peening. While some research has indicated that shot peening does not affect the formation of new phases or compounds [47], others have suggested that as grain refinement increases with shot peening, the number of grain boundaries increases, leading to a subsequent increase in the quantity of unstable phases and compounds [48]. In conclusion, we hypothesize that the presence of XRD peaks for $TiO_2$ and $Al_5Ti_3V_2$ compounds in the shot-peened samples is likely due to the fact that their crystalline content is above the XRD detection limit. This indicates that the crystalline content of these compounds increases with shot peening, which can be attributed to the refinement of grains leading to a greater number of grain boundaries, as previously mentioned [48].

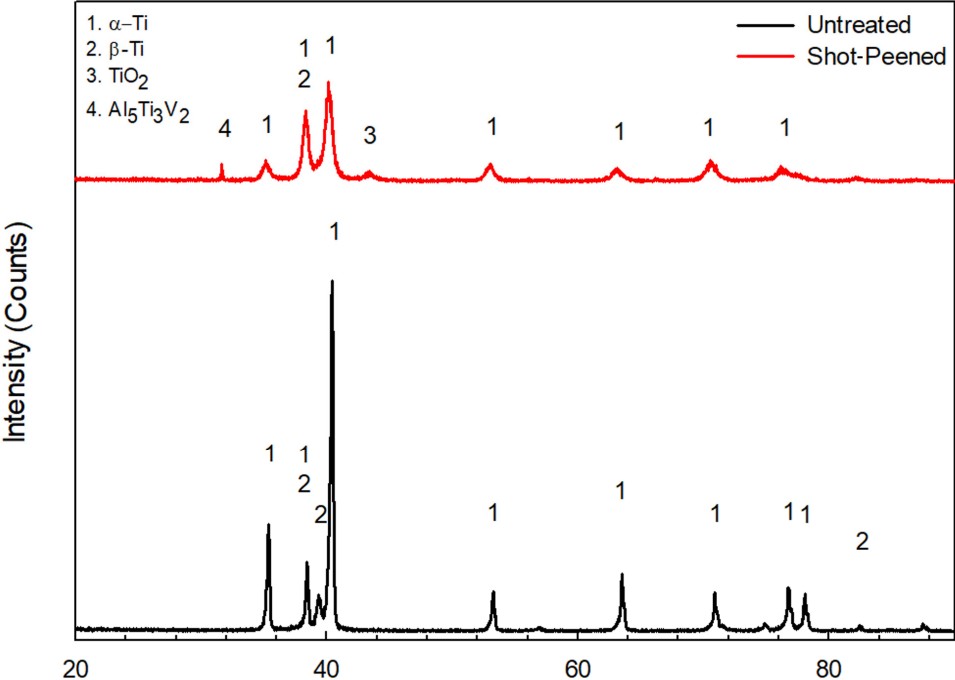

**Figure 13.** XRD patterns of unpeened and shot-peened powder metallurgical Ti6Al4V samples.

The superior corrosion resistance of titanium alloys is due to the dense and thin passive layer formed on the surface [31]. It has been shown that the passive layer depends heavily on the microstructural features (e.g., grain size, grain boundary ratio, amount of dislocation, etc.) [31,49]. The formation and growth of the passive layer on the surface of titanium and its alloys can be found elsewhere [50]. Accordingly, the passive layer is formed via the diffusion of $Ti^{4+}$ and $O^{2-}$ ions from the sample and the corrosive environment to the interface, respectively. Ultimately, the process depends on ion transport (kinetic controlled). The shot peening process may have caused an increase in instability, resulting in an increase in the diffusion of $Ti^{4+}$ ions and a thickening of the passive layer. Comparing the OCP measurements in Figure 6a,b reveals that the shot peening process accelerated the dissolution and diffusion of titanium on the unstable surface. As a consequence, the passive layer developed in a shorter period of time and exhibited steady behavior. In addition, the positive shift of the corrosion potential (from −0.386 V to −0.175 V) indicates that the protection of the passive layer is increased (Figure 7) [51]. The shot-peened sample may have stimulated the formation of a passive zone in the anodic polarization curve, depending on the increase in protection. As previously discussed, shot peening increases the surface area by increasing the roughness. Research has shown that rougher surfaces result in a higher corrosion rate, probably due to the increased surface area [52]. Thus, the higher surface area in contact with the corrosive medium in the shot-peened samples may

detrimentally affect the corrosion resistance, leading to an increase in the corrosion current density ($i_{corr}$, from 1.73 μA to 2.00 μA).

The Bode and Nyquist curves demonstrate that the passive layer produced on the surface of the unpeened Ti6Al4V sample provides excellent protection (Figures 8 and 9). According to the Bode curves, curves for both samples contain a single peak. This shows that a charge-controlled reaction occurs at the corrosive medium–sample interface [29] (Figure 8). In addition to titanium oxide, vanadium oxide also exists in the Ti6Al4V alloy [53]. As mentioned before, the passive layer on the surface of the Ti6Al4V alloy consists of bulk and porous layers, both of which are $TiO_2$-based. According to the EIS results, although the shot peening process did not cause a significant change in the corrosion mechanisms, it significantly increased the protection of both layers. Considering that shot peening notably modifies the microstructural characteristics of the alloy (i.e., grain refinement, an increase in grain boundaries, and dislocation accumulation (Figure 12)), the resulting work-hardened layer (Figure 11) would have an influence on the corrosion behavior of the shot-peened alloy. As previously discussed, the shot-peening-induced changes in surface parameters (roughness, topography, surface area, and morphology) also have an impact on corrosion behavior.

Figure 14 provides a schematic to illustrate the influence of these factors on the corrosion behavior of the shot-peened Ti6Al4V alloy, indicating their negative and positive contributions to the corrosion behavior of the alloy. For the present study, the Bode and Nyquist curves demonstrate that the formation of the corrosion-resistant passive layer dominates the corrosion behavior of the Ti6Al4V alloy; thus, grain refinement (i.e., formation of an ultra-fine grained gradient microstructure) at the surface and sub-surface stimulates the formation of the corrosion-resistant passive layer. Consequently, shot peening notably improved the corrosion resistance of the Ti6Al4V alloy compared to the unpeened samples.

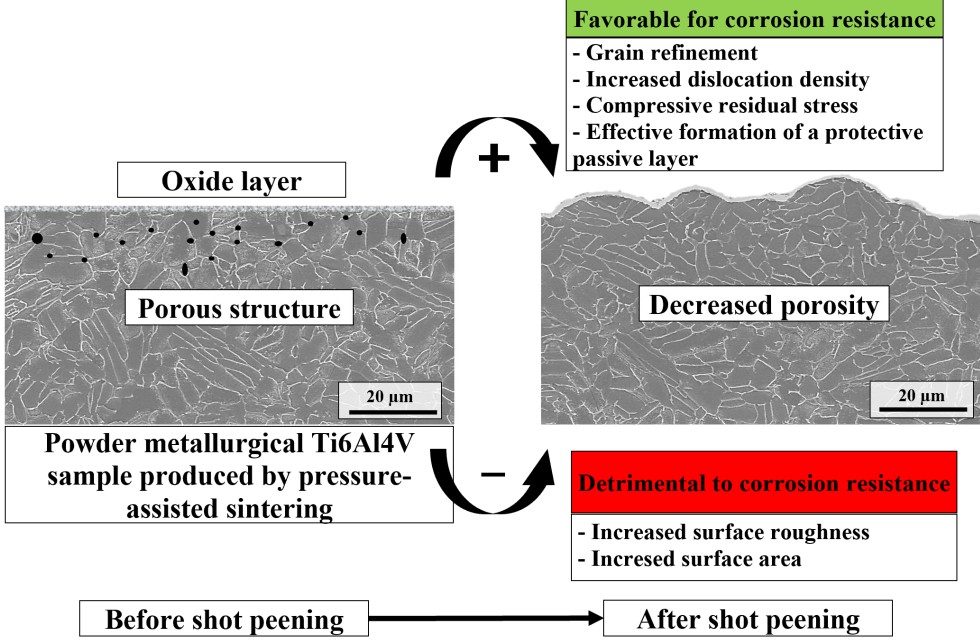

**Figure 14.** Schematic illustration indicating the favorable and detrimental outcomes of the shot peening process with regard to the corrosion resistance of the Ti6Al4V alloy.

## 5. Conclusions

In the present study, the corrosion, surface, and subsurface properties of a Ti6Al4V powder metallurgical alloy produced via pressure-assisted sintering were investigated. Using potentiodynamic polarization and electro-chemical impedance spectroscopy, the effect of shot peening on the corrosion properties was studied. Surface topography, rough-

ness, morphology, subsurface microstructure, and mechanical features were studied to determine the underlying reasons for the changed corrosion behavior with shot peening.

Shot peening produced homogeneous peaks and valleys, with a maximum peak height of 4.83 μm and a peak depth of 6.56 μm. This was attributed to the significant plastic deformation that occurred on the surface because of shot peening. Following the application of shot peening, a 100-micron-thick fine-grained microstructure was formed beneath the surface, and the sizes of the crystallites decreased from 48.59 nm to 27.26 nm. A work-hardened layer with a hardness that increased from 380 Hv to 440 Hv formed beneath the surface down to a depth of 100 μm due to severe plastic deformation and grain refinement.

According to electrochemical impedance spectroscopy, the corrosion resistance of the alloy was enhanced via the formation of a passive film. The effective development of this protective passive layer was supported by the buildup of $TiO_2$ and $Al_5Ti_3V_2$ compounds as a result of shot peening, as determined through XRD analysis. As indicated by the reduction in corrosion potential from $-0.386$ V to $-0.175$ V after shot peening, the passive layer formed more rapidly and was more stable in comparison to the unpeened sample, hence enhancing its resistance to corrosion.

In the present study, for the first time, the effects of shot peening on the corrosion behavior of a powder metallurgical Ti6Al4V alloy produced via pressure-assisted sintering are shown in relation to the modified surface and subsurface properties (roughness, topography, morphology, microstructure, and hardness). The findings will likely contribute to surface engineering and corrosion research, paving the way for the safer and more inventive use of shot peening in applications where corrosion behavior is critical. It is recommended that future research focus on conducting a more comprehensive analysis of pertinent corrosion parameters, including the critical current density, passivation potential, pitting potential, and pitting current density. Additionally, advanced characterization techniques (e.g., transmission electron microscopy) could be utilized to clarify the effects of shot peening on the corrosion performance of Ti6Al4V alloys by elucidating the changes in the corrosion protective passive film's properties (e.g., composition and thickness).

**Author Contributions:** Conceptualization, E.A. (Egemen Avcu), E.A. (Eray Abakay) and Y.Y.A.; methodology; E.A. (Egemen Avcu), Y.Y.A., E.I., İ.G., E.Ç., E.A. (Eray Abakay), F.G.K. and R.Y.; software, E.A. (Egemen Avcu), Y.Y.A., E.I. and E.A. (Eray Abakay); validation, E.A. (Egemen Avcu), Y.Y.A., E.I. and E.A. (Eray Abakay); investigation, E.A. (Egemen Avcu), Y.Y.A., E.I., E.Ç., İ.G., E.A. (Eray Abakay) and F.G.K.; resources, E.A. (Egemen Avcu), A.A., R.Y. and M.G.; data curation, E.A. (Egemen Avcu), Y.Y.A., E.I., E.Ç., İ.G., E.A. (Eray Abakay), F.G.K., R.Y. and M.G.; writing—original draft preparation, E.A. (Egemen Avcu), Y.Y.A., E.I., M.G. and E.A. (Eray Abakay); writing—review and editing, E.A. (Egemen Avcu), E.A. (Eray Abakay), Y.Y.A., E.I., R.Y., A.A. and M.G.; visualization, E.A. (Egemen Avcu), E.A. (Eray Abakay), Y.Y.A. and E.I.; supervision, E.A. (Egemen Avcu), E.A. (Eray Abakay), Y.Y.A., R.Y. and M.G.; project administration, E.A. (Egemen Avcu) and M.G.; funding acquisition, E.A. (Egemen Avcu), A.A., R.Y. and M.G. All authors have read and agreed to the published version of the manuscript.

**Funding:** The authors acknowledge the financial support by Kocaeli University Scientific Research Projects Coordination Unit (Project Number: FCD-2023-3305).

**Institutional Review Board Statement:** Not applicable.

**Informed Consent Statement:** Not applicable.

**Data Availability Statement:** Data contained within the article.

**Conflicts of Interest:** The authors declare no conflict of interest.

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
