# Peer review of "Corrosion Behavior of Shot-Peened Ti6Al4V Alloy Produced via Pressure-Assisted Sintering"

_coatings, doi:10.3390/coatings13122036_

Round 1

Reviewer 1 Report

Comments and Suggestions for Authors

Thank the authors for submitting this paper to the Coatings journal. This work requires significant modifications to be qualified for publication. All the comments are mandatory and must be carefully addressed to avoid delay in the peer-review process or an unfavourable recommendation.

Comment (1): The authors claimed that "The present study, for the first time, investigates the corrosion, surface, and subsurface properties of a shot-peened Ti6Al4V powder-metallurgical alloy.". The reviewer believes the literature does not support this statement. Shot-peening of Ti6Al4V powder-metallurgical alloy is not a novel process. Please refer to the following literature sources as a few examples:

https://doi.org/10.1016/j.jmrt.2021.01.091   

https://doi.org/10.1088/2051-672X/ac4c83    

https://doi.org/10.1016/j.corsci.2021.109558  

Therefore, the following two statements could not be supported: "However, the literature lacks comprehensive research revealing the influence of shot peening on the corrosion behavior of Ti6Al4V alloy, specifically for powder metallurgical Ti6Al4V alloy" and "To the authors' knowledge, the corrosion behavior of powder metallurgical Ti6Al4V alloy has been thoroughly researched for the first time in the present study."

Comment (2): Remove "superficial" from this statement as it is vague: "Thus far, some contradicting and also superficial results have been reported regarding the corrosion behavior of shot peened Ti6Al4V alloy."

Comment (3): Report working and counter electrodes' surface area and shape.

Comment (4): Please specify if EIS was conducted versus OCP. How about a potentiodynamic polarization test? Such essential details must be set.

Comment (5): According to Fig. 6a, a steady-state OCP value was not yet reached in the 1800s. Therefore, the 1800s of OCP monitoring is insufficient to allow the electrochemical interface to get its electrochemical equilibrium condition.  

Comment (6): The term "Tafel curves" is wrong. Those curves are called "potentiodynamic polarization curves."

Comment (7): The authors must specify how the Tafel extrapolation techniques were applied to the curves in Fig. 7.

Comment (8): The authors must discuss if the anodic and cathodic reactions are under kinetic control, mass-transfer control or a mixed control mechanism.

Comment (9): Anodic and cathodic Tafel slope parameters must be reported. The authors must discuss how those Tafel slopes affect corrosion phenomena kinetically.

Comment (10): The authors mentioned the passivation of the peened sample. Based on the shape of the anodic region of the potentiodynamic polarization curve in Fig. 7, the authors must discuss if the behaviour is passivation or pseudo-passivation.

Comment (11): The following statement requires supporting citations: "Due to the passive and transpassive zones formed in the anodic polarization curves, the corrosion rate could not be estimated for either sample."

Comment (12): Extract the following parameters from the polarization curve of peened sample in Fig. 7: passivation potential, critical current density, passivation current density, pitting potential, and pitting current density.

Comment (13): At what potential the EIS test was conducted?  

Comment (14): Add the fitting lines to the EIS curves in Figs. 8 and 9.

Comment (15): Report the equivalent electric circuit simulating the EIS profiles.

Comment (16): Calculate the equivalent electric circuit parameters in a separate Table.

Comment (17): Discuss the physical significance of the circuit model.

Comment (18): Calculate the chi-square and sum of squares parameters. 

Comments on the Quality of English Language

Moderate modification is required, mainly appropriately using "the" and spell checking. 

Author Response

Dear Reviewer#1,

We appreciate your kind comments and suggestions, which have helped us significantly strengthen the manuscript.

We have addressed your comments and suggestions in the revised version of the manuscript. Our responses are given in a point-by-point manner in the attached response letter, with changes to the manuscript highlighted in yellow.

Kind regards,

Authors

Reviewer 2 Report

Comments and Suggestions for Authors

1. The study of this work is interesting. However, it is important to note that the study of corrosion behavior of the shot peened Ti6Al4V alloy is not a new topic. The authors claimed that the novelty of this work is “corrosion behavior of powder metallurgical Ti6Al4V alloy has been thoroughly researched for the first time in the present study”. However, it is worth mentioning that the passive layer, supposed to be with a thickness of 4-6 nm, formed was characterized by SEM rather than TEM. The novelty of the wok should be reconsidered.

2. Do the surface roughness have any correlation with thickness of the passive layer?

3. A more detailed and direct conclusion should be provided.

Comments on the Quality of English Language

OK.

Author Response

Dear Reviewer#2,

We appreciate your kind comments and suggestions, which have helped us significantly strengthen the manuscript.

We have addressed your comments and suggestions in the revised version of the manuscript. Our responses are given in a point-by-point manner in the attached response letter, with changes to the manuscript highlighted in yellow.

Kind regards,

Authors

Reviewer 3 Report

Comments and Suggestions for Authors

The manuscript titled "Corrosion Behavior of Shot Peened Powder Metallurgical Ti6Al4V Alloy" has been reviewed. The comments and suggestions can be found below.

  1. 1. Section Abstract: The abstract should be supplemented with information about the quantitative results obtained.

  2.  
  3. 2. Section Introduction: The sentence "In accordance with the nature of the powder metallurgy (PM) method, the corrosion properties of Ti6Al4V alloy differ from those produced by conventional casting procedures." should be supported by relevant references.

  4.  
  5. 3. Section 2.2: The authors claim, "Surfaces of Ti6Al4V alloy samples were shot-peened using a custom-designed, fully automated shot-peening system; shot peening operating parameters were chosen depending on previous research [1,17,25,26]." While a detailed description of the shot-peening parameters may be available in previous works by the authors, it is important to explain the main criteria behind these choices.

  6.  
  7. 4. Section 2.4: Please include information about the type of hardness tester used. Additionally, specify the number of repeats performed for the hardness test.

  8.  
  9. 5. Section 2.4: The authors claim, "The hardness maps were then visualized using a modified version of a MATLAB® tool code, as reported before [17,27,28]." However, no information about hardness or the MATLAB tool code from the cited paper number 27 was found.

  10.  
  11. 6. Section Discussion: It would be beneficial to discuss the effects of roughness and hardness on the corrosion behavior of Ti6Al4V alloy after shot peening in more detail. Support these discussions with relevant references.

Author Response

Dear Reviewer#3,

We appreciate your kind comments and suggestions, which have helped us significantly strengthen the manuscript.

We have addressed your comments and suggestions in the revised version of the manuscript. Our responses are given in a point-by-point manner in the attached response letter, with changes to the manuscript highlighted in yellow.

Kind regards,

Authors

Reviewer 4 Report

Comments and Suggestions for Authors

- Abstract is too long. Please shorten it.

- "corrosion properties are essential for their use in a variety of engineering applications" Please describe concrete aplications of usage.

- Author should add to text used SEM detector for SEM pictures.

- Please, describe detaily used equipment for all measurements. Microscopes, profilometer, hardness measurement....

- For each measuring descibe norm and also methodology.

- I don't think, that is adequate to express roughness parameters for 3 decimal places. 2 are better.

- Unit should be in brackets "()" in all graphs.

Author Response

Dear Reviewer#4,

We appreciate your kind comments and suggestions, which have helped us significantly strengthen the manuscript.

We have addressed your comments and suggestions in the revised version of the manuscript. Our responses are given in a point-by-point manner in the attached response letter, with changes to the manuscript highlighted in yellow.

Kind regards,

Authors

Round 2

Reviewer 1 Report

Comments and Suggestions for Authors

The paper could be accepted

Reviewer 2 Report

Comments and Suggestions for Authors

None

Comments on the Quality of English Language

None